# Using Image Texture Analysis to Evaluate Soil–Compost Mechanical Mixing in Organic Farms

Elio Romano, Massimo Brambilla *, Carlo Bisaglia and Alberto Assirelli

Council for Agricultural Research and Economics (CREA), Research Centre for Engineering and Agro-Food Processing, Via Milano 43, 24047 Treviglio, BG, Italy; elio.romano@crea.gov.it (E.R.); carlo.bisaglia@crea.gov.it (C.B.); alberto.assirelli@crea.gov.it (A.A.)
* Correspondence: massimo.brambilla@crea.gov.it; Tel.: +39-036349603

**Abstract:** Soil amendments (e.g., compost) require uniform incorporation in the soil profile to benefit plants. However, machines may not mix them uniformly throughout the upper soil layer commonly explored by plant roots. The study focuses on using image texture analysis to determine the level of mixing uniformity in the soil following the passage of two kinds of harrows. A 12.3-megapixel DX-format digital camera acquired images of soil/expanded polystyrene (in the laboratory) and soil/compost mixtures (in field conditions). In the laboratory, pictures captured the soil before and during the simulated progressive mixing of expanded polystyrene particles. In field conditions, images captured the exposed superficial horizons of compost-amended soil after the passage of a combined spike-tooth–disc harrow and a disc harrow. Image texture analysis based on the gray-level co-occurrence matrix calculated the sums of dissimilarity, contrast, entropy, and uniformity metrics. In the laboratory conditions, the progressive mixing resulted in increased image dissimilarity (from $1.15 \pm 0.74 \times 10^6$ to $1.65 \pm 0.52 \times 10^6$) and contrast values (from $2.69 \pm 2.06 \times 10^6$ to $5.67 \pm 1.93\ 10^6$), almost constant entropy ($3.50 \pm 0.25 \times 10^6$), and decreased image uniformity (from $6.65 \pm 0.31 \times 10^5$ to $4.49 \pm 1.36 \times 10^5$). Using a tooth-disc harrow in the open field resulted in higher dissimilarity, contrast, entropy (+73.3%, +62.8%, +16.3%), and lower image uniformity (−50.6%) than the disc harrow, suggesting enhanced mixing in the superficial layer.

**Keywords:** GLCM; soil organic matter; image dissimilarity; image contrast; image entropy; image uniformity; harrowing





## 1. Introduction

Soil is a complex medium consisting of minerals, organic matter, micro-organisms, air, and water whose physical, chemical, and biological characteristics mainly result from the interaction of the solid components with the vertical water flow [1]. Such features make soil an essential, non-renewable resource that supports, regulates, and provides agricultural ecosystems [2]. To provide the best environment for plant roots, soil amendments (SA) improve and maintain soil physical properties, i.e., water retention, permeability, infiltration, drainage, aeration, and structure. Soil quality is strictly related to its structure. Much of the environmental damage to intensively farmed lands (e.g., erosion, compaction, and desertification) originates from soil structure degradation that may result from agricultural practices. Long-term cultivation lowers soil organic matter (SOM) content, whereas fertilization, the input of SA (i.e., manure and compost), and fallow commonly enhance its content [3]. Animal manure is the most common SA: its land application maximizes its agricultural value, minimizing its potential impact on environmental quality and human health [4]. At the same time, biowaste compost or compost-derived products represent valid SAs for stockless and vegetable farms (and also run organically). Without them, such farms can hardly meet the non-leguminous grain crops' N demand and sustain soil humus with only organic sources [5–9]. The nutrient dynamics in the soil are closely linked

to biologically active SOM resulting from either recent organic matter inputs or accumulated soil reserves [10]. For this reason, following the quality of the procurement, besides the evenness of the distribution, the compost mixing accuracy in the upper layer of soil deserves attention too.

The mechanized processes for administering organic fertilizers are essential to soil fertility retention: machines shall incorporate solid organic fertilizers uniformly through the profile and comply with the varying needs of the soil plots. However, there are additional requirements that SA-distributing machinery should comply with: e.g., safety, versatility, compliance with the fertilization regulatory framework, uniform product distribution, the ability to work on both horizontal and sloping surfaces, and variable-rate administration of the SA [11–13]. The SA-spreading operations always imply a uniformity spreading error; however, a distribution pattern presenting a coefficient of variation lower than 20% makes such an error acceptable [14]. Many studies have focused on the distribution uniformity of solid-fertilizer-spreading machinery using a weight-based approach. Among these, Vasilica et al. [15] analyzed four possible situations focusing on combining maximum uniformity with a minimum distribution by collecting the distributed material on a 1 m$^2$ surface and weighting it afterwards. Using the same approach, other researchers studied the distribution pattern of variously formulated organic fertilizers and SAs [16,17]. Some studies focused on the mechanical aspect of the distribution machinery, designing devices and mechanical systems to transport, meter, and spread organic fertilizers uniformly [18,19].

Following the introduction of image analysis techniques, researchers adopted an approach for the characterization of the soil pore system [20], the study of the distribution of plant nutrients in soil cores [21], and the characterization of the particles of organic and inorganic fertilizers [22,23]. Furthermore, image analysis also proved to allow the automatic detection of the spread granules of fertilizer [24].

More in detail, the image texture features have the advantages of considering visual characteristics that do not depend on image color or brightness and providing reference to the homogeneous phenomenon of the image (i.e., they describe the pixel distribution in light of their neighborhood space) [25–27]. Furthermore, when focusing on a given image detail, texture features contain information about the captured surface structure arrangement, reflecting its connection with the surrounding environment [28,29].

The gray-level co-occurrence matrix (GLCM) is the most common image texture analysis method [30–36] because it reflects all the possible information within a grayscale image, e.g., direction, interval, amplitude, and change ratio. The GLCM is a tabulation of how often different combinations of pixel brightness values (grey levels) occur in an image object in a given direction. It reveals specific attributes about the gray-level spatial distribution in an image object, allowing the derivation of statistical indices (metrics) that Hall-Beyer [37] separated into three groups: (i) the "contrast group", which includes contrast, dissimilarity, and uniformity; (ii) the "orderliness group", which includes entropy, angular second moment, and energy; and (iii) the "descriptive statistics group" that relates to mean, variance, standard deviation, and correlation, calculated using the entries in the GLCM, not the original pixel values. GLCM texture feature extraction occurs when analyzing the local features of an image for pattern recognition, image classification, and image segmentation [38–45]. This approach has been rarely used in the field of soil science; however, some studies tested the feasibility of image texture features from GLCM to determine the correlation between soil moisture conditions and the intensity of the pixel in laboratory conditions [46] while, in the open field, they tested image texture parameters from Sentinel-1 for soil moisture retrieval [47]. Recently, Zhao et al. [48] used the GLCM texture analysis to describe the surface cracking conditions of soda saline–alkali soil and quantitatively studied the responses of GLCM texture features to soil salinity.

This work introduces the adoption of image texture features (i.e., dissimilarity, contrast, entropy, and uniformity) to evaluate the uniformity of mixing of a composted SA in the upper soil layer following the passage of two kinds of harrow in organic vegetable farms. The hypothesis is that the information resulting from their dynamics relates to the

level of mixing that different harrows induce in SA in the upper soil layer, following the appropriateness that texture metrics showed for mapping changes even in situations with complex structures, such as forests with understories or mixed forests [49–52] or images and composite mosaic datasets of a coral reef [53]. The present study foresaw a preliminary tuning of the method In laboratory conditions to test the image texture metric' behavior at increasing levels of particle dispersion in soil. At this study stage, the experimental activity foresaw the use of expanded polystyrene particles (EPS) to mimic and visualize the dispersion of SA particles when subjected to harrowing. The acquired information was afterwards tested in field conditions. Such activity occurred within a research project on soil tilling in organic horticultural sowing seed production.

The innovation of this study relies upon the possibility of a quick check of the mixing level that different harrows achieve in the upper soil layer to exploit at their best the amending properties of SAs [5–10] and, following their adequate mixing in the upper soil level, reduce the uncertainty deriving from the heterogeneity of their materials to comply with precision compost strategies [54] in the framework of precision agriculture. This becomes particularly important in Southern Europe, where the Mediterranean climate and land use are responsible for steady organic matter depletion [55,56].

## 2. Materials and Methods

### 2.1. Compost Distribution and Soil Mixing

Spreading tests occurred in two organic farms (labelled *Carpinello* and *Ponticelli*) located in an agricultural district of East Po Valley (Emilia Romagna), which was classified as "under desertification" at the end of the 1990s [55]. The cultivated fields are for the production of sowing seeds for horticulture. Both make use of massive spreading of green composted SA (i.e., higher than 23 t ha$^{-1}$ y$^{-1}$) purchased from the same manufacturer (Enomondo Srl, Faenza, Italy). Tables 1 and 2 report the main soil features of the sites and the main characteristics of the used SA.

**Table 1.** Main soil features for the upper 0–0.3 m layer in the considered sites [57,58].

| Soil Characteristics | *Carpinello* | *Ponticelli* |
|:---:|:---:|:---:|
| Sand (%) | 5.31 | 25.52 |
| Silt (%) | 47.59 | 51.68 |
| Clay (%) | 47.10 | 22.80 |
| SOM (g $kg_{soil}^{-1}$) | 1.66 | 2.04 |
| P$_2$O$_5$ (g $kg_{soil}^{-1}$) | 32.0 | 42.0 |
| K$_2$O (g $kg_{soil}^{-1}$) | 542.0 | 159.0 |
| pH | 8.0 | 7.7 |

**Table 2.** Main characteristics of the green composted SA (data from manufacturer).

| SA Characteristics | Average Content Range |
|:---:|:---:|
| Moisture (%) | 22–32 |
| pH | 6.5–7.5 |
| Organic carbon (% d.m.) [1] | 22–26 |
| Humic and fulvic carbon (% d.m.) | 6.0–8.0 |
| Total N (% d.m.) | 1.2–1.8 |
| Organic N (% d.m.) | 1.2–1.8 |
| C/N ratio | 15–19 |
| Salinity (meq/100 $g_{d.m.}$) | 19–52 |
| P (% d.m. as P$_2$O$_5$) | 0.4–0.6 |
| K (% d.m. as K$_2$O) | 1.0–1.2 |

[1] d.m. = dry matter.

Compost distribution and spreading occurred in both farms using a three-axis spreader wagon (manufactured by Serri s.n.c, Predappio, Italy) trailed by a four-wheel-drive tractor.

The spreader equipment included a distribution system with counter-rotating basal plates and a punctual adjustment system with precise adjustment of the SA distribution rate. The distribution of such a high dose of SA on the surface gave rise to a thick layer of soil amendment.

After distribution, in one farm (labelled *Carpinello* farm), SA incorporation into soil took place using a combined spike-tooth–disc harrow; in the other, marked *Ponticelli* farm, SA incorporation foresaw the use of a disc harrow. Both machines had a mixing depth lower than 15 cm.

### 2.2. Image Acquisition

Image acquisition occurred in both organic farms with a 12.3-megapixel DX-format NIKON D300 digital camera (Nikon Corporation, Minato, Tokyo, Japan) 42 days after compost distribution following Ortiz et al.'s [59] recommendations.

For each sampling site, digging occurred in three different places of the compost-amended fields to expose the soil's superficial layer and acquire three images in each place for the subsequent image processing (nine images for each site).

### 2.3. Tuning of the Method in Laboratory Conditions

Before processing the open field captured images, an artificial soil profile was simulated using a glass case of 403 mm × 238 mm × 497 mm to test the method's power in laboratory conditions. The case was initially filled with 8 mm sieved soil to simulate the superficial horizon. Afterwards, expanded polystyrene (EPS) particles were spread on the surface (15 mm thick layer) to emulate compost distribution, and a small shovel simulated the mixing action of the disc harrow executing four passages. EPS particles were used to obtain information on the GLCM metrics dynamics because EPS particle color profoundly differs from the soil color, giving rise to pictures containing well-defined visual edges (meaning clear-cut changes between EPS and soil particles and, therefore, neighboring pixels). Next, three pictures of the profiles were taken on the three sides of the case after each mixing action using the same digital camera described in Section 2.2. for a whole thirty-six pictures. Image capturing occurred before polystyrene distribution and after each of the three consecutive mixings that occurred afterwards. Finally, the resulting digital images underwent image processing (Section 2.4). This test aimed to check the discrimination power of the method and gain insights into the meaning of the calculated metrics regarding SA mixing with soil.

### 2.4. Image Processing

Each picture of the soil sections underwent processing with the R-4.3.0 statistical software [60]: The raster function of the raster package [61] read the picture space composed of cells of equal size (pixels—units of the coordinate reference system). Subsequently, a rectangle including the upper 100 mm of the soil profile (without sky) cropped from the picture (Figure 1) underwent further analysis to create a GLCM, generally used in texture analysis because it captures the spatial dependence of gray-level values within an image [62]. This second-order statistic algorithm, included in the GLCM package for R, compares two neighboring pixels simultaneously to point out how often a pixel with *i* intensity (gray-level) occurs in a specific spatial relationship to a pixel with the value j within a restricted area [63].

In a few words, each element (*i,j*) in the resultant GLCM is the frequency at which the pixel with a value *I* occurred in the specified spatial relationship to a pixel with value *j* in the original image. Such processing allowed the calculation of four features on the GLCM-processed images [64,65]: dissimilarity, contrast, entropy, and uniformity. Figure 2 reports a visual example of the processing that the images underwent.

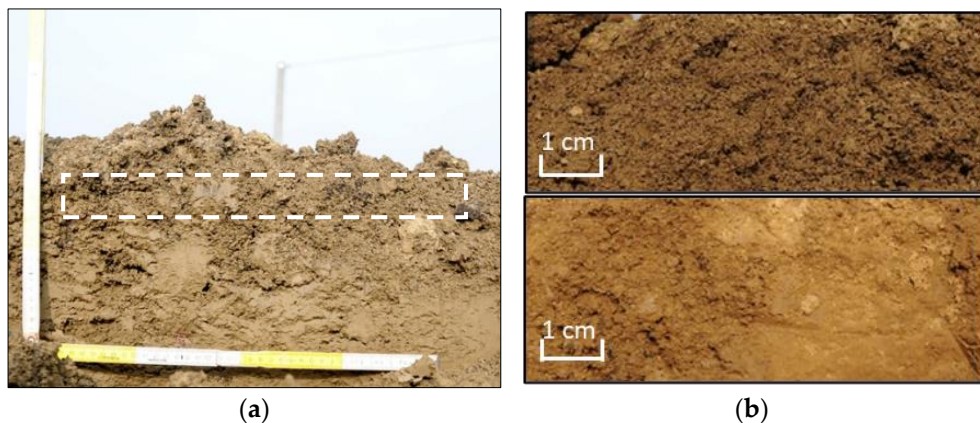

(a)            (b)

**Figure 1.** Soil mixed with compost in field conditions: (**a**) example of image acquisition and image cropping for GLCM analysis; (**b**) examples of cropped images taken after the passage of a combined spike-tooth–disc harrow (*Carpinello* farm, **above**) and a disc harrow (*Ponticelli* farm, **below**).

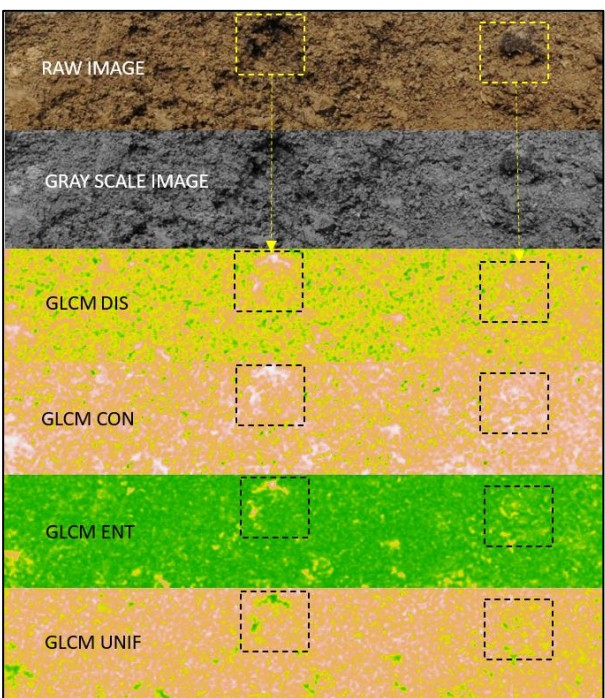

**Figure 2.** Picture showing the progress of GLCM image processing from the raw image (top of the figure) to the processed images resulting for each considered metric. The dashed squares refer to the compost particles dispersed into the soil profile.

Dissimilarity (DIS): A measure of the distance between pairs of objects (pixels) in the region of interest (Equation (1)). It indicates how far apart the values of neighboring points on the surface are: low values represent remarkable homogeneity.

$$\text{DIS} = \sum_{i,j=0}^{N-1} P_{i,j} |i-j| \tag{1}$$

Contrast (CON): This statistic measures an image's spatial frequency. It results from the difference between the highest and the lowest values of a contiguous set of pixels resulting in the number of local variations in the image. It represents the amount of local gray-level variation in an image. A high value of this parameter may indicate the presence of edges, noise, or wrinkled textures. A low-contrast image presents the GLCM

concentration term around the principal diagonal and features low spatial frequencies (Equation (2)).

$$\text{CON} = \sum_{i,j=0}^{N-1} P_{i,j}(i-j)^2 \tag{2}$$

Entropy (ENT): This statistic (Equation (3)) measures the disorder or complexity of an image. The entropy is high when the image is not texturally uniform, and many GLCM elements have minimal values. When complex, textures tend to have high entropy: overall, it gives the reason for the randomness, having its highest value when the elements of an analyzed surface are all equal.

$$\text{ENT} = \sum_{i,j=0}^{N-1} P_{i,j}\left(-\ln P_{i,j}\right) \tag{3}$$

Uniformity (UNIF) or homogeneity measures the uniformity (or orderliness) of the gray-level distribution of the image: images with a smaller number of gray levels have more considerable uniformity (Equation (4)).

$$\text{UNIF} = \sum_{i,j=0}^{N-1} P(i,j)^2 \tag{4}$$

In all the equations, $P_{ij}$ is the element $i,j$ of the normalized symmetrical GLCM, and $N$ is the number of gray levels in the image. For each metric, data processing calculated the sum, the average, the median, and the maximum values for the cropped rectangle. Then, based on the size of the cropped rectangle of soil (Figure 1), the sum of the values was calculated for each metric.

The processing results were the average sums of the metrics resulting from the replicates (nine for each side of the case for the laboratory activity and nine pictures for each field site).

## 3. Results

Figure 3 shows how, in laboratory conditions, the progressive mixing of EPS particles with soil (a small shovel simulated the passage of the concave metal disc of the harrow) resulted in a progressively more dispersed EPS particle redistribution in the upper profile.

On the other hand, image entropy (ENT) remains almost constant. At the same time, image uniformity (UNIF) tends to decrease, meaning that subsequent mixing actions result in slight image texture changes and decreased image uniformity following the dispersion of the EPS particles in the upper layer. These results confirmed the expectations of laboratory-induced mixing: the metrics follow the progressive redistribution of EPS particles in the soil profile. Concerning the identification of the achieved level of mixing, UNIF and DIS have the highest efficiency, and such metrics are almost uncorrelated between themselves (r = −0.07). At the same time, UNIF is moderately and negatively correlated with CONT (r = 0.25) and positively and moderately correlated with ENT (r = 0.36).

Figure 4 shows the indices related to such images expressed as boxplots of the sums of the metrics. During the progressive mixing of the EPS particles, the sums of the DIS and CON indices show an increase, meaning that the digital images move from low to higher spatial frequencies.

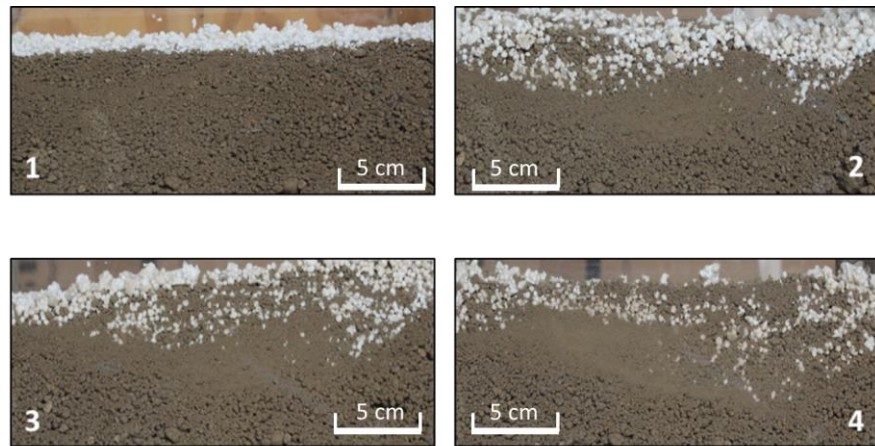

**Figure 3.** Examples of the pictures of the soil profiles from the laboratory test. Image acquisitions represent the initial state with the layer of EPS before mixing (**1**) and after the first three simulated mixings (from **2** to **4**).

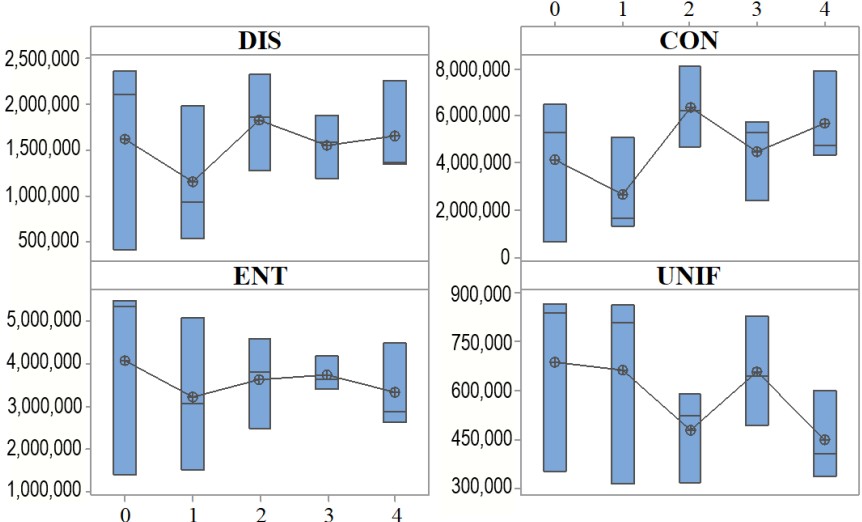

**Figure 4.** Boxplot representation of the sum of image texture metrics without any EPS on the surface (0), with the EPS layer before mixing (1), and after each mixing action (2–4).

Figure 5 represents the output of the processing of the indices resulting from the GCLM analysis on the images from the *Carpinello* and *Ponticelli* farms.

After distributing the SA with the same device, the action of different harrows results in upper soil layers with varying image textures. Based on the studied metrics, the passage of a combined spike-tooth–disc harrow (*Carpinello* farm) results in more dispersed compost particles than a disc harrow (*Ponticelli* farm). Figure 4 shows such variations: the sums of DIS, CON, and ENT show a significant decrease ($p < 0.05$), meaning that, in the region of interest, the distance between pixels in the *Ponticelli* soil is lower than in *Carpinello*. On the contrary, the UNIF index increased, albeit non-significantly, suggesting the presence of fewer gray levels in *Carpinello* than *Ponticelli* soil, which, as abovementioned, is ascribable to the existence of more coarse particles of compost in the second farm site.

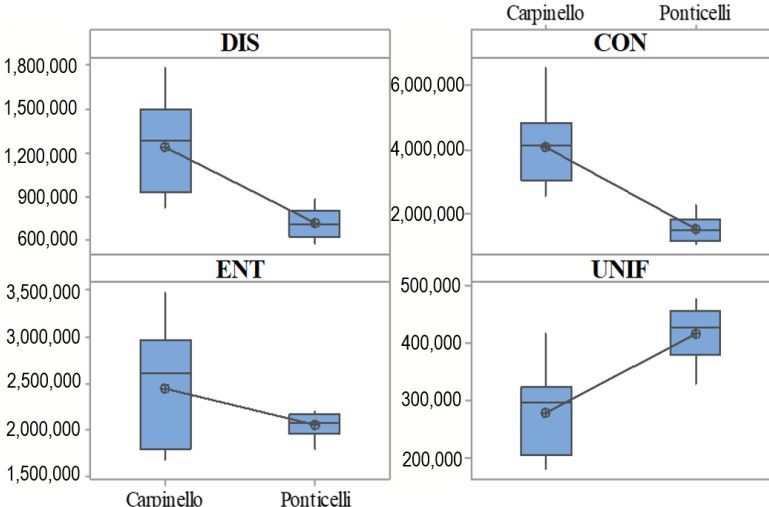

**Figure 5.** Boxplots representing the sums of the considered indices in each sampling site. The boxplot contains the median and the mean value: a line connects the latter.

## 4. Discussion

This study focuses on setting up a straightforward image-based methodology to assess the differences between these two kinds of harrows. Harrows are alternative tillage implements used for minimum tillage. They cut the soil to a shallow depth for smoothening and pulverizing it, cutting the weeds, and mixing the materials on the surface with the soil. The disc harrow operates through a single set or multiple sets of rotating discs mounted on a common shaft. The discs rotate on the ground as the tractor pulls the harrow ahead, cutting the lumps of soil, clods, and roots and mixing the material throughout the first layer of soil. In addition, some disc harrow models may be equipped with horizontal bars carrying straight teeth (combined spike-tooth–disc harrow) to further smooth and level the ploughed soil or the seedbed before planting or sowing. The result is a smooth field with powdery dirt at the surface whose structure is open and homogeneous, allowing better water movement, particularly when harrowing follows the distribution of compost or manure [66].

As the touch recognizes various objects according to their tactile texture, the tangible feel of a surface, in image processing, texture (meant as the set of metrics describing the spatial variation in pixels' brightness intensity) is the primary term used to define objects or concepts of a given image [67]. Therefore, image texture analysis is essential in computer vision cases (e.g., object and pattern recognition, surface defect detection, and medical image analysis). Moreover, image texture is one of the most powerful methods for classifying or segmenting an image [68,69]. With remote sensing techniques, such an approach proved to be feasible for classifying land use and improving the recognition of crop early phenological stages using machine learning algorithms [70–72]. In proximal sensing applications, it allowed recognition of the human skin as an indication of the presence of people (human limbs or torso) within a digital image [73] or to identify cells from damaged and intact tissue in histologic images [74].

In this study, the test in laboratory conditions aimed to determine to what extent and how the considered indices (image features) vary with the progressive mixing of the upper layer of soil. Under the mixing action of the small shovel, the increasing dispersion of EPS particles gives rise to images that move towards a growing level of uniformity (Figure 3). The variation that the calculated image features show for each mixing step (Figure 4) is remarkable for DIS, CON, and UNIF, but ENT is the measure that changed the least. According to Hall-Beyer [41], ENT might be able to characterize a particular image section; however, it might also take on different values from varying edges' characteristics. Hall Beyer [75] referred to DIS, CON, and ENT as "edge textures". These yield high values when the neighborhood contains abrupt color changes between neighboring pixels,

which have some spatial coherence to contrasting pixel pairs. Higher ENT values result for neighborhoods containing very irregular edges or incoherent contrast, whereas straight-line edges would lower ENT values for the neighborhood.

In this case, the ENT value does not change significantly from picture 1 to picture 4, meaning that the captured complexity does not change significantly and that the matrix elements are almost equal [33] following a large amount of uniform soil compared to the EPS particles. DIS and CON follow the same dynamics under the findings of Hall-Beyer [75]; moreover, the increasing values indicate high variations in the gray level of image matrices, which means the texture becomes irregular following the dispersion of EPS particles. On the other hand, such particle redistribution causes UNIF to drop because the edge indicating the cut-off line between the soil and EPS particles fades.

When applying such achievements to the images taken in the organic fields after SA distribution, the metrics point out that the pictures taken in the Ponticelli site have more constant pixels than *Carpinello* (Figure 5). Moreover, the significantly lower value of ENT ($p < 0.05$) provides insights that the pixels of the image taken in *Ponticelli* are significantly more texturally uniform, meaning that the used machinery caused SA to be more dispersed in the soil profile.

The desired aggregate size of soils in seedbeds varies because of crop-specific requirements. However, in practice, soil conditions for seedbed preparation are mainly based on farmers' qualitative field assessment, which relies on observing the breaking of soil aggregates. Although farmers can carry out the qualitative assessment of soil with fair precision, the results are subjective because the method is intuitive and, therefore, operator-dependent [76]. Oduma et al. [77] pointed out that the soil type affects the performance of the implementation, reporting harrowing field efficiencies of 85.83% for loamy sandy soils (such as *Ponticelli*) and 84.95% for clay loam soil (such as *Carpinello*). Such a difference in field efficiency may explain the improved mixing that image texture metrics suggest for the *Ponticelli* site.

Concerning the machinery, the adopted harrows are pretty widespread in the organic farms of the region; the main difference relies upon the different forces the soil particles undergo when varying the functional elements of the harrow. On the one hand, the only discs of the disc harrow operate mainly a horizontal displacement of the soil particles: first outwards to the working section and then conveying them back towards the inner part of the working section. On the other, the presence of the vertical elements determines deeper cracks throughout the profile, which allow a more profound mixing of the SA with the soil particles.

## 5. Conclusions

The study presents an image characterization of amended soil pictures resulting from SA distribution using a combined spike-tooth–disc harrow (Carpinello farm) and a disc harrow (Ponticelli farm) to evaluate the possibility of using image texture analysis to assess the uniformity of distribution of the soil amendment through the upper horizon.

A laboratory-scale experiment pointed out the dynamics of four texture metrics (i.e., dissimilarity, contrast, entropy, and uniformity) at increasing levels of dispersion of EPS particles, mimicking the behavior of SA particles in the soil.

The results of this study indicate that the GLCM approach is an effective method for evaluating the dispersion of the compost particles added onto soil and afterwards dispersed in the surface layer under the action of the harrows. The image texture metrics successfully evaluated the changes occurring in the morphology and surfaces of the EPS particles increasingly dispersed through the upper horizon of the soil, and provided helpful hints to infer the level of SA dispersion in the studied sites resulting from the different used harrows. In addition, the metric dynamics indicate that developing an image evaluation tool can be important for targeting the SA dispersion, thus improving the efficiency of the added organic matter. Based on the processing results, the action of a combined spike-tooth–disc harrow results in better SA mixing with soil particles than a disc harrow.

Further studies (e.g., aimed at the automatic recognition of shades of gray corresponding to the compost particles) are, however, needed to widen the applicability of the tested method and include it in a machine learning algorithm for the automated recognition of the mixing level that a machinery achieves.

**Author Contributions:** Conceptualization, A.A., M.B., E.R. and C.B.; methodology, M.B. and E.R.; formal analysis and data processing, M.B. and E.R.; investigation, M.B. and A.A.; writing—original draft preparation, M.B. and E.R.; writing—review and editing, A.A. and C.B.; supervision, C.B.; project administration, A.A.; funding acquisition, A.A. All authors have read and agreed to the published version of the manuscript.

**Funding:** This research was funded by the Italian Ministry of Agriculture, Food and Forestry Policies (MIPAAF): projects AGROENER, grant n. 26329/2016, and INSOBTEC-DIBIO, grant n. 76381 of 31 October 2018.

**Institutional Review Board Statement:** Not applicable.

**Data Availability Statement:** Not applicable.

**Acknowledgments:** The authors acknowledge the help of Gianluigi Rozzoni and Ivan Carminati (CREA, Research Centre for Engineering and Agro-Food Processing) in the setup of the method's testing in laboratory conditions.

**Conflicts of Interest:** The authors declare no conflict of interest.

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
