# Peer review of "Using Image Texture Analysis to Evaluate Soil–Compost Mechanical Mixing in Organic Farms"

_agriculture, doi:10.3390/agriculture13061113_

Round 1
Reviewer 1 Report
Comments are attached.

Author Response
Dear Reviewer,
I have attached the pdf file containing the response to your comments and remarks.
Regards

Reviewer 2 Report
Dear Author,
The study was not carried out under real conditions. This is the major shortcoming of this study. You can find detailed some comments in the orijinal paper. Therefore, the research is rejected.
Best wishes.
Author Response
Dear reviewer,
I am sorry. We probably could not be sufficiently clear in the description of the test, which was carried out in laboratory conditions for the tuning of the method and the open field for the testing. The revised manuscript contains more details that may help the evaluation.
Regards
Reviewer 3 Report
OBSERVATIONS TO MANUSCRIPT 2380482: Using image texture analysis to evaluate soil-compost mechanical mixing in organic farms.
The document is interesting, it presents a methodology that allows to observe the distribution of the compost. The results showed, in the laboratory conditions, the progressive mixing resulted in increased image dissimilarity and contrast values, almost constant entropy and decreased values of image uniformity. In the open field, using a spike tooth-disk harrow resulted in higher dissimilarity, contrast, entropy, and lower image uniformity than the disc harrow, suggesting improved mixing in the superficial layer.
The writing can improve if the observations are considered. There are parts of the writing that are taken for granted; however, you must be more specific. It is important that you try to clarify the questions that were raised and redraft where you are asked to do so.
Introduction
Line 63. It's written: … fertilizers and Sas [16- 17]. … . Please, the sentence must be written: … fertilizers and Sas [16,17]. … .
Line 65-66. It's written: … uniformly [18-19]. … . Please, the sentence must be written: … uniformly [18,19].
Another important aspect that was not observed in the introduction, the authors did not make an assumption or hypothesis about the behavior of the relationship between the variables in this study. This situation is important, as it would allow readers to identify the scope of the investigation.
Materials and Methods
Line 81. It's written: … with massive (i.e. higher than 23 t ha-1 y-1 ) green composted SA spreading. … . Please, the sentence must be written: … with massive spreading of green compost (i.e. more than 23 T ha-1 ere applied per year) as soil amendments.
Line 82-90. What are the characteristics of the compost from both places?
Line 102. Why did you use polystyrene? What characteristics does this material have that allowed it to resemble compost? Was it because of the shape, size, weight, density, color?
Line 113. Why was R software used to divide the image into pixels of the same size, if the image is made up of pixels, which gives the image the resolution?
How did you ensure the same scale in the images (Styrofoam image and field images), regardless of pixelation?
How do you ensure that pixel i or j are organic particles or aggregates? This question is asked because the basic principles of image texture analysis is the supervised classification of the image, based on prior knowledge, in this case on the gray scale of organic matter.
Results
It would be interesting if the authors showed the natural and grayscale image of the distribution of organic matter, in order to indicate the pixels that represent the compost aggregates in the soil.
Line 243. You did not use the gray level co-occurrence matrix to evaluate the soil texture. You evaluated the dispersion of the compost particles that was added and dispersed in the surface layer. Please, write again.
Line 252. I agree, but before making this recommendation, you could consider classifying the shade of gray that corresponds to the compost particles (coarse or fine), in order to identify them after they are mixed into the soil and subsequently carry out automatic recognition.
Author Response
Dear Reviewer,
Please find attached the pdf file containing the response to your comments and remarks.
Regards

Reviewer 4 Report
This manuscript evaluates the efficiency of soil-compost mechanical mixing in organic farms through the textural attributes derived from the gray-level co-occurrence matrix. Lab and field photos from a digital camera were obtained to derive four textural features: dissimilarity, contrast, entropy, and uniformity. I have the following comments for the revised version of the manuscript:
1. The possible differences in the soil chemical and physical properties of the two farms may have affected the conclusion that the spike tooth-disk harrow is better than the disc harrow to mix composted soils. As the ideal condition of having both types of mechanization on the same farm is not always possible, some words about this possible difference in the soil types needs to be addressed in the manuscript.
2. In the Introduction section, I suggest the authors highlight some of the most common applications of the GLCM and then state that this approach has been rarely used in the field of Soil Science.
3. Some soil physical data from the two farms should be presented by the authors, especially those related to soil color, soil moisture condition during the photo shooting, and soil texture.
4. The photos obtained by the lab experiment using expanded polystyrene (EPS) are not so convincing. Photos 2 and 3 are quite similar and photo 4 is not so different from the others. This may be the reason why entropy did not present statistical differences. If possible, I recommend repeating this experiment using much more contrasting conditions.
5. Figure 2 shows four conditions named 1, 2, 3, and 4 while Figure 3 shows five conditions named 0, 1, 2, 3, and 4. What is the 0 condition?
6. The authors said that three images were obtained from two farms. This strategy generates 9 (3 images x 3 bands) GLMC features per farm and per textural attribute. How were they converted from nine to single GLCM features?
7. Please, add bar scales to the photos shown in Figures 1 and 2.
8. I believe there is still room to provide representative figures of GLCM attributes so that the readers can have a better idea about the performance of each textural feature.
English writting is fine.
Author Response
Dear Reviewer,
I am sharing with you the pdf file containing the response to your comments and remarks.
Regards

Round 2
Reviewer 1 Report
I have checked the revision and authors' response to my comments, it is acceptable at present format.
Author Response
Dear Sir or Madam,
Thank you for your acceptance of the manuscript.
Regards
Reviewer 2 Report
The paper can be accepted
Best wishes
Author Response

(The authors gave the same response as above.)

Reviewer 3 Report
OBSERVATIONS TO MANUSCRIPT 2380482 (2): Using image texture analysis to evaluate soil-compost mechanical mixing in organic farms.
Responses to comments are accepted.
However, I make two more suggestions:
- line 73-89. For your research, what are dynamics metrics and how do you justify their use in your work?
- When you mention polystyrene, please write the justification for its use, as you wrote in your response to the observation, and complement why its use can simulate the movement of organic particles in the soil.
Author Response
Dear Sir or Madam,
I have attached the response to your remarks and suggestions.
Regards

Reviewer 4 Report
All my previous concerns were addressed properly so that I consider this revised version of the manuscript ready to be published. Congratulations to the authors. In the next submissions by the authors, I recommend highlighting the changes made in the revised versions of the manuscript in red color, regardless of selected journal. This is a common procedure for other authors.
English writting is fine.
Author Response
Dear Sir or Madam,
Thank you for your acceptance of the manuscript.
I apologize for the colour of the revisions: now, the red colour has been set as a reference.
Regards